# Association between car driving and successful ageing. A cross sectional study on the "S.AGES" cohort

Edouard Baudouin[1,2]☯*, Sarah Zitoun[1,3]☯, Emmanuelle Corruble[1,3], Jean-Sébastien Vidal[4,5], Laurent Becquemont[1,6], Emmanuelle Duron[1,2]

1 Université Paris-Saclay, UVSQ, CESP, Team MOODS, Le Kremlin-Bicêtre, France, 2 Département de Gériatrie, Assistance Publique—Hôpitaux de Paris, Hôpital Paul Brousse, Villejuif, France, 3 Département de Psychiatrie, Assistance Publique—Hôpitaux de Paris, Hôpital Bicêtre, Le Kremlin-Bicêtre, France, 4 Département de Gériatrie, Assistance Publique—Hôpitaux de Paris, Hôpital Broca, Paris, France, 5 Université Paris Descartes, INSERM, Paris, France, 6 Clinical Research Center, Assistance Publique—Hôpitaux de Paris, Hôpital Bicêtre, Le Kremlin-Bicêtre, France

☯ These authors contributed equally to this work.
* edouardpierre.baudouin@aphp.fr

**Data Availability Statement:** Qualified researchers may request access to patient level data and related study documents including the clinical study report, study protocol with any amendments, blank

## Abstract

### Background

Assessing successful ageing (SA) is essential to identify modifiable factors in order to enforce health promotion and prevention actions. SA comprises 3 dimensions: an active engagement with life, a low probability of disease and disease-related disability, and a high cognitive and physical functional capacity. Driving seems to be linked to SA as it is a mean to preserve social interactions and requires preserved functional and cognitive status. This study aims to investigate whether driving status can be considered a proxy of SA, by describing determinants associated with driving status in the 65+.

### Methods

This cross-sectional study is ancillary to the S.AGES (Sujets AGÉS—Aged Subjects) study, an observational prospective cohort study which included patients suffering from chronic pain, type-2 diabetes mellitus or atrial fibrillation from 2009 to 2014. SA was defined by the success of three dimensions: physiological comprised of comorbidity and autonomy scores, psychological comprised of cognitive status and emotional state, and a social dimension.

### Results

2,098 patients were included of whom 1,226 (58.4%) reported being drivers. 351/2,092 (16.7%) were classified as successful agers: 292/1,266 (23.8%) in the driver group vs. 59/872 (6.8%) in the non-driver group; p < .001. In the final logistic model, after adjustment for relevant variables, SA was associated with driver status OR 1.94 [1.36–2.77].

case report form, statistical analysis plan, and dataset specifications. Patient level data will be anonymized and study documents will be redacted to protect the privacy of our trial participants. Further details on Sanofi's data sharing criteria, eligible studies, and process for requesting access can be found at: https://www.vivli.org/.

**Funding:** This study was supported by SANOFI France The funders had no role in study design, data collection and analysis, decision to publish, or preparation of the manuscript.

**Competing interests:** L.B appeared/appears as principal or co-investigator of several clinical trials from different sponsors: Novartis, Actelion, Alnylam Pharmaceuticals, IONIS Pharmaceutical, Takeda, Novonordisk, Shire, Boehringer. This does not alter our adherence to PLOS ONE policies on sharing data and materials.

## Conclusion

Driving may be considered as a proxy to SA: it reflects elders' independence, cognitive ability and a means to maintain social interactions. To preserve their mobility and enable them to achieve SA, regular screening of driving skills, specific rehabilitation programs are needed. Moreover development and communication on special transports services, communal rides or even driverless car to avoid apprehension around older adults driving could be solutions.

## Introduction

Lifespan is increasing spectacularly since the beginning of the industrial era. In France in 2018, life expectancy was 85.3 years and 79.4 years for females and males respectively [1]. Thus, participation of older persons in road traffic is also increasing: in 2019, out of 549 subjects aged from 86 to 101 years old, 16% were regular car drivers [2]; in the European Union, by 2030, a quarter of licensed drivers will be aged 65 and older [3]. However, the 65+ years old (y/o) represent a risk regarding road safety: according to the French National observatory of road safety in 2021, 13% of the 2,994 road fatalities were 65 y/o or more [4]. This is mainly due to comorbidities and cognitive and sensory impairments [5]. Nonetheless, driving is also a major factor for our elders' social interactions as shown in a prospective cohort of 4,359 community-dwelling older adults (mean age 78.72 (7.32) y/o): frequent drivers are more likely to visit friends and family (odds ratio (OR) 1.75; $p < .01$), go out for entertainment (1.75; $p < .001$), attend to religious services (1.77; $p < .01$) compared with subjects who ceased driving [6]. However, driving represents a complex activity as it implies multiple simultaneous tasks with different temporal and cognitive requirements [7]. Finally, it requires one to be physically fit as it requires good eyesight, coordination, strength and muscle control [8].

These three aspects (social, cognitive and functional) are parts of successful aging (SA) as theorized by Rowe and Kahn, who defined it as: an active engagement in life, a low probability of disease and disease-related disability, and a high cognitive, physical and functional capacity [9]. in a large cohort of 2475 subjects, it has been shown that drivers had higher SA score (OR 0.65 [0.54–0.77]) [10].

However, it has been pointed out in a systematic review [11] that the association between driving and successful ageing is still poorly understood and that modes of transports are insufficiently studied with SA.

Therefore, the aim of this study was to investigate whether driving status can be considered a proxy of SA, by describing driving status determinants in the 65+.

## Material and methods

### Study design, setting, and participants

This cross-sectional study is ancillary to the S.AGES (Sujets AGÉS—Aged Subjects) study. This report follows the STROBE recommendations (S1 Checklist).

The S.AGES study was an observational prospective cohort study from 2009 to 2014. The main objective was to describe therapeutic management of outpatients. Inclusion criteria were being 65 years and older and suffering from chronic pain (n = 1400), or type-2 diabetes mellitus (n = 1,004), or atrial fibrillation (AF; n = 1,087) which defined the 3 sub-cohorts. Six hundred and sixty French general practitioners (GP) were randomized into one of the sub-cohorts

and were asked to include 1/3 of participants aged 65–75 years old, and 2/3 of participants aged of 75 years and older. Participants were assessed by their GP every 6 months for three years. Socio-demographic, clinical and treatment data were recorded at inclusion, and updated at each visit except for driving status. Polypharmacy was defined as taking 5 or more treatments, physiological age was defined as GPs' assessment of patients' physiological age as less than, equal to or greater than their chronological age [12].

The non-inclusion criteria were institutionalization, inability to give consent or to take part in the study follow-up, participation in another clinical trial and presence of a life-threatening disease with less than3 months of life expectancy. All patients gave and signed an informed consent to participate in the study. All procedures were in accordance with institutional guidelines and approved by the local ethics committee (Comité de protection des personnes Ile de France XI) on January 15, 2009 (ref 09006) and the French National Agency for Medicines and Health Products (ANSM) on February 6, 2009 (ref B81333-40) (ClinicalTrials.gov NCT01065909). Full methodology and characteristics of the cohort have already been published [13].

## Outcome

SA was defined according to Young et al definition [14] which includes physiological, psychological and social dimensions. However, because the cohort was not specifically developed to answer all exact items of Young et al definition, some proxies were used (S1 Table).

- **Physiological component.**

- The comorbidity dimension was constructed as a continuous variable by summing 16 frequent chronic comorbidities in elders: history of stroke, heart disease (atrial fibrillation, valvulopathy, presence of pacemaker or implantable cardioverter-defibrillator, congestive heart failure), peripheral arterial disease, venous thromboembolism, hypertension, Parkinson disease, thyroid dysfunction, type-2 diabetes, osteoarthritis, osteoporosis, rheumatoid arthritis, chronic pain, cancer, liver disease (cytolysis or cirrhosis), chronic respiratory disease (chronic obstructive pulmonary disease, sleep apnea, fibrosis), peptic ulcer. In order to be as conservative as possible, having no more than 3 of the aforementioned diseases [15] was considered a success.

- Autonomy dimension was measured with Activity of Daily Living (ADL; scored out of 6: bathing, dressing, toilet hygiene, transferring, self-feeding, continence) [16], Instrumental Activities of Daily Living (IADL; scored out of 4: use of telephone, use of mean of transport, drug management, finance management) [17], presence of professional caregiver and a fall in the past 12 months. Success was defined by respectively a score of 6/6 for the ADL, 4/4 for the IADL, the absence of a care taker and the absence of falls in the last 12 months.

- Physiological component was deemed as success if the comorbidity and autonomy dimensions were successful.

- **Psychological components.**

- Cognitive function was assessed by Mini Mental State Evaluation (MMSE) [18]. Success was defined as scores equal or greater than to 27 (maximum score 30),

- Depressive state was assessed by the Geriatric Depressive Scale (GDS [19]. A risk of depressive state was defined with a score of at least 10 [20]. (Maximum score 15).

Psychological component was a success if both cognitive function and depressive state were successful.

• **Social component.**   Only one social component could be assessed with the study data: social isolation defined by the living condition (alone or not). This measure, however not complete, is still relevant as social isolation is a major risk factor for elders in term of morbidity and mortality [21, 22].

The aging process of the subjects was deemed successful if all 3 components (physiological, psychological and social) were successful.

### Statistical analysis

Data are presented with mean and standard deviation (SD) for continuous variables and count (percentage) for categorical variables. Normal distribution was assessed graphically for each continuous variable. Comparisons were made with t-tests for continuous variables and chi-squared tests or Fisher's exact tests for categorical variables. Patients' characteristics were described overall and according to the driving status. Finally, a logistic regression was performed with the driving status as the dependent variable and the independent variables chosen with a conservative approach ($p < .10$ on the univariate analysis, and all clinically relevant variables from the literature). Variables included in the SA model were excluded in order to avoid multicollinearity; correlation was assessed by a focused principal component analysis [23]. In case of visual proximity, the most clinically relevant variable was kept. Then, a stepwise logistic regression was performed to select the final model, and multiple correlation bias was assessed by the variance inflation factors [24]. Results are presented with odds ratios and 95% confidence interval (CI). Validity conditions of the logistic regression model were also assessed.

The distribution of missing values was calculated in the two groups (drivers and non-drivers). Patients with missing values on driving status, cognitive status, and depression status were excluded. Thus, a direct comparison between subjects with significant missing values and non-missing values was made (S2 Table). Missing values of the relevant variables of the multivariate model represented less than 5% of the overall data available and were missing at random, which allowed us to perform multiple imputations by chained equations based on a Monte-Carlo Markov Chain [25]. Ten multiple imputations were used with 20 maximum iterations; binary data were imputed by logistic regression, qualitative variable by polynomial regression and quantitative variable by predictive mean matching [26].

Analyses were performed with R V4.0.0.

### Results

From the 3,491 subjects included in the original cohort, 82 were excluded due to un-matched inclusion criteria and 1,336 were excluded because of missing values. Thus, assessment of subjects with missing values was performed (S2 Table). These missing values were not missing at random: they were mainly present in subjects who did not drive. 2,098 patients were included at first visit with complete information on driving status, GDS and MMSE (Fig 1). Finally, because only 6 subjects had responded "Currently working" regarding their professional activity, they were merged with the retired subjects as opposed to the "never had a professional activity" group.

Among them, 1,226 (58.4%) declared to be drivers, mean age (SD) 79.4 (6.2) vs. 75.6 (5.8) y/o in the non-driver group $p < .001$, a physiological age more frequently equal to or greater than their chronological age 1146 (93.5%) vs. 763 (87.5%); $p < .001$ and a mean MMSE of 27.9 (2.4) vs. 26.7 (2.9) in the non-driver group; $p < .001$. Demographic and clinical data of the population are presented in Table 1, heart diseases, pulmonary and liver condition details are given in **S3 Table**.

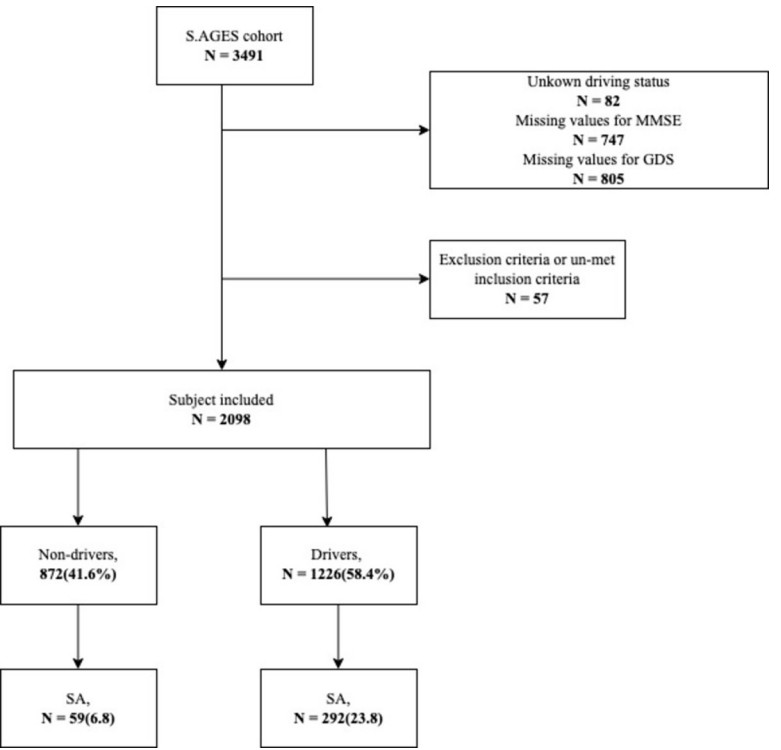

**Fig 1. Flow chart.**

Regarding SA, mean age was 74.4 (5.5) y/o (vs. 77.9 (6.2); p < 0.001), 238 (67.8%) were male (vs. 644 (39%); p <0.001). Overall, 351 (16.7%) were classified as successful agers: 292 (23.8%) in the driver group vs. 59 (6.8%) in the non-driver group; p <0.001:

- Among the drivers, 535 (43.6%) were successful in the physiological component: 673 (54.9%) reached the comorbidity component and 898 (73.2%) reached the autonomy component. In the non-drivers, 166 (19.0%) were successful in the physiological component (p < .001 compared with drivers) with 356 (40.8%) successful in the comorbidity component (p < .001) and 311 (35.7%) successful on the autonomy component (p < .001)

- 884 (72.1%) achieved the psychological component in the drivers vs. 451 (51.7%) in the non-drivers (p< .001),

- 912 (74.9%) succeeded in the social component in the drivers vs. 464 (53.2%) in the non-drivers (p < .001).

Confusion matrix and parameters in term of sensitivity, specificity and area under the curve can be found in S1 File Confusion matrix of SA and Driving status.

In the final logistic model, after adjustment for age, sex, physiological age, education level, current work status, living in urban or rural area, alcohol and tobacco consumption and polypharmacy, SA was associated with driving status 1.94 [1.36–2.77] (Table 2).

## Discussion

This analysis found that driving status tended to be associated with SA (OR 1.94 [1.36–2.77]) after adjustment for age, sex, physiological age, education level, current work status, living in urban or rural area, alcohol and tobacco consumption and polypharmacy.

**Table 1. Descriptive and univariate analysis according to driving status.**

| | Total N = 2098 | Non-drivers 872 (41.6%) | Drivers 1226 (58.4%) | P value |
|---|---|---|---|---|
| **Socio-demographic parameters n (%)** | | | | |
| Observatory | | | | |
| Chronic pain | 857 (40.8) | 439 (50.3) | 418 (34.1) | < .001 |
| Atrial fibrillation | 636 (30.3) | 226 (25.9) | 410 (33.4) | |
| Type 2 diabetes | 605 (28.8) | 207 (23.7) | 398 (33.4) | |
| Age, mean (SD) | 77.2 (6.2) | 79.4 (6.2) | 75.6 (5.8) | < .001 |
| Physiological age | | | | |
| Less than chronological age | 512 (24.4) | 211 (24.2) | 301 (24.6) | < .001 |
| Equal to chronological age | 1397 (66.6) | 552 (63.3) | 845 (68.9) | |
| Greater then chronological age | 186 (8.9) | 109 (12.5) | 77 (6.3) | |
| Missing values | 3 (0.1) | 0 (0) | 3 (0.2) | |
| Sex, Females | 1170 (55.8) | 737 (84.5) | 433 (35.3) | < .001 |
| ADL[a], mean (SD) | 5.9 (0.4) | 5.7 (0.6) | 5.9 (0.2) | < .001 |
| Missing values | 5 (0.2) | 1 (0.1) | 4 (0.3) | |
| IADL[b], mean (SD) | 3.7 (0.7) | 3.3 (0.9) | 3.9 (0.4) | < .001 |
| Missing values | 10 (0.5) | 6 (0.7) | 4 (0.3) | |
| Education level | | | | |
| Primary school | 872 (41.6) | 452 (51.8) | 420 (34.3) | < .001 |
| Secondary school | 772 (36.8) | 287 (32.9) | 485 (39.6) | |
| High school | 237 (11.3) | 74 (8.5) | 163 (13.3) | |
| University | 192 (9.2) | 50 (5.7) | 142 (11.6) | |
| Missing values | 25 (1.2) | 9 (1) | 16 (1.3) | |
| Professional status | | | | < .001 |
| Currently working or retired | 1749 (83.4) | 622 (71.3) | 1127 (91.9) | |
| Missing values | 15 (0.7) | 7 (0.8) | 8 (0.7) | |
| Alcohol consumption | 553 (26.4) | 103 (11.8) | 450 (36.7) | < .001 |
| Missing values | 18 (0.9) | 5 (0.6) | 13 (1.1) | |
| Tobacco consumption | | | | |
| Never | 1524 (72.6) | 756 (86.7) | 768 (62.6) | < .001 |
| Former | 499 (23.8) | 90 (10.3) | 409 (33.4) | |
| Current | 64 (3.1) | 20 (2.3) | 44 (3.6) | |
| Missing values | 11 (0.5) | 6 (0.7) | 5 (0.4) | |
| Living area | | | | |
| Rural | 502 (23.9) | 160 (18.3) | 342 (27.9) | < .001 |
| Semi-rural | 530 (25.3) | 177 (20.3) | 353 (28.8) | |
| Urban | 1066 (50.8) | 535 (61.4) | 531 (43.3) | |
| Residency | | | | |
| Alone at home | 716 (34.1) | 408 (46.8) | 308 (25.1) | < .001 |
| Not alone at home | 1351 (64.4) | 438 (50.2) | 913 (74.5) | |
| Living facility | 31 (1.5) | 26 (3) | 5 (0.4) | |
| Professional caregiver | 413 (19.7) | 286 (32.8) | 127 (10.4) | < .001 |
| Missing values | 53 (2.5) | 16 (1.8) | 37 (3) | |
| Polypharmacy[c] | 1284 (61.2) | 570 (65.4) | 714 (58.2) | < .001 |
| Comorbidities sum, mean (SD) | 3.6 (1.5) | 3.9 (1.5) | 3.4 (1.5) | < .001 |
| Missing values | 41 (2) | 13 (1.5) | 28 (2.3) | |
| Alzheimer's disease | 397 (18.9) | 159 (18.2) | 238 (19.4) | 0.496 |

*(Continued)*

**Table 1.** (Continued)

| | Total N = 2098 | Non-drivers 872 (41.6%) | Drivers 1226 (58.4%) | P value |
|---|---|---|---|---|
| | | | | |
| Missing values | 5 (0.2) | 2 (0.2) | 3 (0.2) | |
| Osteoarthritis | 1118 (53.3) | 553 (63.4) | 565 (46.1) | < .001 |
| Missing values | 2 (0.1) | 0 (0) | 2 (0.2) | |
| Cancer | 267 (12.7) | 87 (10) | 180 (14.7) | *0.001* |
| Missing values | 4 (0.2) | 1 (0.1) | 3 (0.2) | |
| Chronic pain | 1352 (64.4) | 650 (74.5) | 702 (57.3) | < .001 |
| Fall in the past year | 209 (10) | 132 (15.1) | 77 (6.3) | < .001 |
| Missing values | 7 (0.3) | 1 (0.1) | 6 (0.5) | |
| Peptic ulcer history | 66 (3.1) | 23 (2.6) | 43 (3.5) | 0.259 |
| Missing values | 6 (0.3) | 2 (0.2) | 4 (0.3) | |
| GDS[d] < 10 / 15 | 201 (9.6) | 120 (13.8) | 81 (6.6) | < .001 |
| Heart condition | 932 (44.4) | 365 (41.9) | 567 (46.2) | 0.035 |
| Missing values | 14 (0.7) | 3 (0.3) | 11 (0.9) | |
| Hypertension | 1479 (70.5) | 645 (74) | 834 (68) | *0.004* |
| Missing values | 4 (0.2) | 1 (0.1) | 3 (0.2) | |
| MMSE[e] ≥ 27 / 30 | 1461 (69.6) | 514 (58.9) | 947 (77.2) | < .001 |
| Missing values | 1 (0) | 0 (0) | 1 (0.1) | |
| Liver disease | 36 (1.7) | 13 (1.5) | 23 (1.9) | 0.503 |
| Missing values | 5 (0.2) | 2 (0.2) | 3 (0.2) | |
| Osteoporosis | 300 (14.3) | 202 (23.2) | 98 (8) | < .001 |
| Missing values | 10 (0.5) | 5 (0.6) | 5 (0.4) | |
| Type II diabetes | 851 (40.6) | 325 (37.3) | 526 (42.9) | 0.011 |
| Missing values | 2 (0.1) | 2 (0.2) | 0 (0) | |
| Thyroid dysfunction | 268 (12.8) | 134 (15.4) | 134 (10.9) | *0.003* |
| Missing values | 19 (0.9) | 7 (0.8) | 12 (1) | |
| Parkinson disease | 28 (1.3) | 18 (2.1) | 10 (0.8) | 0.014 |
| Missing values | 4 (0.2) | 0 (0) | 4 (0.3) | |
| Pulmonary condition | 254 (12.1) | 95 (10.9) | 159 (13) | 0.148 |
| Missing values | 4 (0.2) | 1 (0.1) | 3 (0.2) | |
| Rheumatoid arthritis | 106 (5.1) | 50 (5.7) | 56 (4.6) | 0.238 |
| Missing values | 5 (0.2) | 0 (0) | 5 (0.4) | |
| Stroke | 46 (2.2) | 26 (3) | 20 (1.6) | 0.038 |
| Missing values | 6 (0.3) | 2 (0.2) | 4 (0.3) | |
| Thromboembolic history | 142 (6.8) | 79 (9.1) | 63 (5.1) | < .001 |
| Missing values | 19 (0.9) | 4 (0.5) | 15 (1.2) | |
| **Successful ageing** | **351 (16.7)** | **59 (6.8)** | **292 (23.8)** | **< .001** |
| Missing values | 96 (4.6) | 34 (3.9) | 62 (5.1) | |
| Physiological component | 701 (33.4) | 166 (19) | 535 (43.6) | < .001 |
| Missing values | 96 (4.6) | 34 (3.9) | 62 (5.1) | |
| Psychological component | 1335 (63.6) | 451 (51.7) | 884 (72.1) | < .001 |

(*Continued*)

**Table 1.** (Continued)

|  | Total N = 2098 | Non-drivers 872 (41.6%) | Drivers 1226 (58.4%) | P value |
|---|---|---|---|---|
| Social component | 1382 (65.9) | 464 (53.2) | 918 (74.9) | < .001 |

Note. Data are number (%) unless otherwise indicated. In case of no missing value, the line empty was kept empty.

[a]ADL = Activities of daily living,

[b]IADL = Instrumental activities of daily,

[c]Polypharmacy ≥ 5 treatments,

[d]GDS = Geriatric Depressive Scale,

[e]MMSE = Mini-Mental State Examination,

Regarding the medico-demographic factors, drivers were significantly younger (OR 0.89 [0.87–0.91]) and had a lower physiological age (OR 0.25 [0.17–0.38]) than non-drivers. This is in line with the high prevalence of comorbidities found in the non-driving group, which were found all significant except for pulmonary, rheumatic diseases, peptic ulcer, peripheral arterial disease and liver diseases. This result also correlates with the high rates of the non-poly-medicated subjects found in the driver group: 714 (58.2%) vs. 570 (65.4%) in the non-driving group (p < .001). However this association was found non-significant in the multivariate analysis probably due to the adjustment for SA. Higher rates of comorbidity were found in other cohorts [27] which is assumed to be related to underdiagnosis in this study. This raises a

**Table 2.** Multivariate analysis of the driving status according to successful ageing adjusted for age, sex, physiological age, education level, professional status, living area, alcohol consumption, tobacco consumption and polypharmacy.

|  | Odds Ratios | CI | p |
|---|---|---|---|
| Successful aging (ref = no) | 1.94 | 1.36 – 2.77 | < .001 |
| Age | 0.89 | 0.87 – 0.91 | < .001 |
| Sex (ref = F) | 6.84 | 5.10 – 9.17 | < .001 |
| Education level (ref = primary school) |  |  |  |
| Secondary school | 2.08 | 1.62 – 2.62 | < .001 |
| High school | 2.95 | 2.00 – 4.37 | < .001 |
| University | 2.79 | 1.79 – 4.35 | < .001 |
| Professional status (ref = never employed) | 1.62 | 1.20–2.20 | < .001 |
| Living area (ref = urban) |  |  |  |
| Rural | 3.00 | 2.25 – 4.01 | < .001 |
| Semi-rural | 2.19 | 1.66 – 2.89 | < .001 |
| Alcohol consumption (ref = no) | 1.68 | 1.24 – 2.28 | < .001 |
| Polypharmacy [a] | 0.87 | 0.69 – 1.10 | 0.260 |
| Physiological age (ref = equal to chronological age) |  |  |  |
| Less than chronological age | 1.33 | 1.02 – 1.74 | 0.03 |
| Greater than chronological age | 0.25 | 0.17 – 0.38 | < .001 |
| Tobacco consumption (ref = never) |  |  |  |
| Former | 1.51 | 1.09 – 2.10 | .001 |
| Current | 1.01 | 0.52 – 1.95 | 0.980 |
| Observations | 2098 |  |  |
| Number of imputations | 10 |  |  |

Notes: OR = Odds Ratio, [a]polypharmacy ≥ 5 treatments

question of GPs' awareness of the comprehensive geriatric assessment, a "technique for multi-dimensional diagnosis of frail elderly people" [28].

As for the current living area, because of a lack of public transportation as stated by a French government report [29], this work found that drivers were more often living in a rural or semi-rural area compared with urban area (OR 3.00 [2.25–4.01] and 2.19 [1.66–2.89]) respectively. Indeed, access is harder in these area (longer distances to bus stops, fewer transports) and car driving is preferred [30]. Regarding gender, male gender was significantly associated with driving (OR 6.84 [5.10–9.17]). This might be explained by the fact that fewer women learnt to drive before 1955, women experience higher rates of different physical and psychological conditions that may lead to driving cessation: osteoporosis, osteoarthritis, pain, urinary incontinence, fear of falling and poorer self-confidence at driving [31]. These results are also consistent with literature on SA [28, 32], where men are more frequently classified as successful agers irrespective of the SA definition used. Alcohol and tobacco consumption were significantly associated with driving status: (OR 1.68 [1.24–2.28] and 1.51 [1.09–2.10]) respectively; their relation to SA is still under discussion in the literature where alcohol (daily consumption) and tobacco consumption are either a positive or negative SA factors [33].

Nonetheless, drivers had a higher probability to be classified as successful agers (OR 1.94 [1.36–2.77]). This can be explained by the necessity to be free of cognitive and functional impairments to be able to perform the complex task of driving [34]. However, the physiological aspect was the component with the lowest success: 701 (33.4%) overall. After reaching a certain age, only few people are free from any conditions. In a cohort of 5820 Japanese American cohort followed for 40 years, only 11% were free of one of 6 major chronic diseases at 85 years old [35]. Secondly, the overall rates of the social component was 1382 (65%), which may be explained by the unique aspect of social interactions evaluated in this study. However, the rates for community dwelling older adults ranged from 87.8% [36] to 24.4% [37], which is explained by the difficulty to measure this dimension. Finally, the results for the psychological component, which referred to both emotional state and cognitive status, were as expected: 1335 (63.6%) subjects were found to be cognitively unimpaired and at low risk of depression [38].

The high rate of subjects who did not reach the psychological component among drivers can be noted. When suffering from dementia, mild cognitive impairment or depression, subjects may frequently have impaired attention and executive functions, or be at higher risk of using drugs that may impair driving ability [39–41]. This result may be explained by the fact that French legislation does not require regular reevaluation of driving abilities. However, we may see improvement in future cohort. Indeed, since the fall of 2022, when suffering from a specific group of diseases (including cognitive impairment), it is required for the patient to undertake a specific medical examination focused on driving abilities by a certified physician. However, this specific medical appointment can only be made by the patient himself after receiving information on the risks and obligations regarding his condition, as medical conditions cannot be disclosed by any health care professional.

Nonetheless, regarding these results, in order to maintain social interactions and to reach SA, we advocate for our elder's mobility. Indeed, it has been shown that driving correlates well with health-related quality of life (HRQOL) in 544 subjects of 90.3 years (± 2.7) on self-care and usual-activities domains (OR 0.41 (95%-CI 0.17 to 0.98 OR 0.48 (0.26 to 0.90) respectively) [42]. This may be achieved either by improving preventive medicine and comprehensive geriatric assessment in order to improve physiological and psychological components. Moreover, development of tools to screen for drivers with cognitive impairment is needed such as Screen for the Identification of Cognitively Impaired Medically At-Risk Drivers or Trail Making Test which have shown high rate of misclassification on pass/fail in a road test among drivers [43, 44]. It is however interesting to note that according to a Canadian study of 108 older adults

(mean age 80.6 +/- 4.9 y/o) on self-awareness of driving performance, 53% over-estimated themselves (vs. 19% underestimated themselves). In the case of irreversible conditions, instead of developing common public transport, development of special transport services for subjects who need special care and communication of their existence (only half of respondents of a large Norwegian survey knew about these services [30]), or maybe the use of driverless cars in association with public transports [45] might be solutions for the impaired 65+.

This study presents some limitations. First, this study concerned the 65 year and older adults suffering from a chronic disease (AF, chronic pain or type 2 diabetes) with regular follow up with a physician. However, this study is still relevant as these three conditions are of high prevalence in Europe and worldwide, especially in the 65+ population [46–48]. Second, we acknowledge that two dimensions may be superficially explored: cognitive impairment and social dimension. Regarding the former, the sole use of MMSE might underestimate cognitive impairment, however widely used as a screening tool [49]. We also recognize that the definition used for social dimension may limit the scope of our results as it only includes living alone or not; which may not always result in loneliness. Nonetheless, this proxy has been shown correlated with other SA definitions [50, 51]. Finally, in order to perform state of the art statistical analysis and to keep a geriatric perspective, some subjects had to be excluded, mainly due to missing values of MMSE and GDS scores. This might be due to the fact that subjects cognitively impaired are less likely to take part in these tests [52]. The analysis showed that these subjects were older, had lower ADL/IADL scores, and had lower success on social or physiological components (S2 Table). Thus, this bias might have impacted on the magnitude of the results but not the conclusions, as the excluded subjects are more likely to be non-drivers. Regardless, this study was composed of a large real-life sample of 65+ y/o with extensive socio-demographical, clinical and therapeutic data.

## Conclusion and implications

The issue of driving in older adults is a major public health topic as it seems to be related to SA. Given the limitations of this study, driving might be considered a proxy of SA: it reveals their autonomy, psychological capability and a way to maintain social interactions. Road safety might be improved by regular screening of driving skills and perhaps specific rehabilitation programs (driving simulator for instance), which may also improve cognitive functions. Nonetheless, mobility needs be preserved for those 65+ in order to reach SA which can be achieved by the development and communication of special transport services, communal rides or even driverless car to avoid apprehension around older adults driving.

## Supporting information

**S1 Checklist. Checklist of items that should be included in reports of cohort studies.**
(DOC)

**S1 Table. Successful ageing definition: Difference between Young et al proposal and study definition.**
(DOC)

**S2 Table. Missing values analysis: MMSE and or GDS missing subjects vs. non MMSE nor GDS values.**
(DOC)

**S3 Table. Heart diseases, pulmonary and liver condition details.**
(DOC)

**S1 File. Confusion matrix of SA and driving status.**
(DOCX)

## Author Contributions

**Conceptualization:** Edouard Baudouin, Sarah Zitoun, Laurent Becquemont,
Emmanuelle Duron.

**Formal analysis:** Edouard Baudouin.

**Funding acquisition:** Laurent Becquemont.

**Methodology:** Edouard Baudouin, Sarah Zitoun, Laurent Becquemont.

**Supervision:** Emmanuelle Duron.

**Writing – original draft:** Edouard Baudouin, Sarah Zitoun.

**Writing – review & editing:** Emmanuelle Corruble, Jean-Sébastien Vidal,
Laurent Becquemont, Emmanuelle Duron.

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
