## [Decision Letter · Decision Letter 0]

28 Nov 2022

PONE-D-22-26799Association between car driving and successful ageing. A cross sectional study on the "S.AGES" cohortPLOS ONE

Dear Dr. Baudouin,

Thank you for submitting your manuscript to PLOS ONE. After careful consideration, we feel that it has merit but does not fully meet PLOS ONE’s publication criteria as it currently stands. Therefore, we invite you to submit a revised version of the manuscript that addresses the points raised during the review process.

We look forward to receiving your revised manuscript.

Kind regards,

Masaki Mogi

Academic Editor

PLOS ONE

Journal Requirements:

   "I have read the journal's policy and the authors of this manuscript have the following competing interests: L.B appeared/appears as principal or co-investigator of several clinical trials from different sponsors: Novartis, Actelion, Alnylam Pharmaceuticals, IONIS Pharmaceutical, Takeda, Novonordisk, Shire, Boehringer"

Additional Editor Comments:

Major revisions are needed in the present form. 

See the Reviewers' comments and respond to them appropriately.

Reviewers' comments:

Reviewer's Responses to Questions

**Comments to the Author**

1. Is the manuscript technically sound, and do the data support the conclusions?

Reviewer #1: Yes

Reviewer #2: Yes

Reviewer #3: Yes

2. Has the statistical analysis been performed appropriately and rigorously? 

Reviewer #1: Yes

Reviewer #2: Yes

Reviewer #3: Yes

3. Have the authors made all data underlying the findings in their manuscript fully available?

Reviewer #1: Yes

Reviewer #2: Yes

Reviewer #3: Yes

4. Is the manuscript presented in an intelligible fashion and written in standard English?

Reviewer #1: Yes

Reviewer #2: Yes

Reviewer #3: Yes

5. Review Comments to the Author

Reviewer #1: This study shows that older drivers are more often successful agers than nondrivers. The finding is clear but not unexpected.

Major concerns:

1. The patients suffer from one of 3 diseases, which should be mentioned already here and also in the title, which suggests that the study deals with healthy elderly.

2. Driving behavior and success are not included, just driving vs. nondriving.

Details:

l.30: unclear, what determinants? No driving determinants are measures, only driving or not driving. The simple finding is that drivers are more often successful agers than nondrivers.

L42: better: it reflects

l.74: determinants? better:... driving status.

l.106: AF was the condition of one of the 3 groups

l.123: the MMSE is a very superficial and unspecific measure for the various cognitive functions. It is not suitable to measure cognition in healthy elderly and is more appropriate to check for MCI or dementia.

l.229: some text is missing after “any”.

l.249: here more possibilities should be mentioned to improve driving capabilities of older drivers, such as physical and cognitive training, as well as driving training.

Reviewer #2: Dear authors,

I have made below a point by point revisions of your interesting manuscript. I think the aim is really interesting but that the manuscript should be brought forward more clearly onto how this will transfer to the aging of drivers, or non-drivers. I'm sure that with this in mind, that the manuscript will be suitable for publication.

In the abstract, percentage should be given with the associated ratio since not of the data presented match in terms of calculations and percentage shown.

Line 51: participation of older persons in traffic should be presented as a prevalence and not older drivers generating traffic on the road.

Line 53: Results from France would be better than the USA to highlight the relative risk of older drivers even though these results might dated.

Line 55: there is an error with the percentage sign (%)

Line 88: how is physiological age defined (and reference) ? Since it is one of the main metric, it has to be better defined herein.

In Table 1, with the age of the sample, how can we be sure that the high percentage of women who are not driving (84.5%) is actually associated with health or social status while they might have never had a driver license from the beginning? It is not uncommon at the age to see older women who never drove and therefore make it difficult to account for driving as a successful indicator of SA if one might have never done it before. This is raised on line 215. For line 216, this sentence should be supported by your results and not the one from the reference 29.

Why is professional status merged for “Currently working or retired”?

Line 205 : the vs needs non-driving group after for clarity.

Line 242 (presented as 442 in the response to reviewers) lacks scientific support in this domain in regards of evaluation and screening for at risk drivers and would merit a more detailed discussion (See Brenda Vrjklan’s work or Michel Bedard for more details).

Line 211 raises an important issue, if driving seems closely link with urban vs rural area, why would not cessation of driving be seen as SA if one is voluntarily transitioning from car ownership to public and active transportation? You refer to it on line 269. Your argument should be consistent.

Line 246 (not Line 446): Public transport and its associated services cannot be placed in contradiction with autonomous vehicles. The latest have been sold has the new panacea for older drivers for the last 10 years despite showing real improvement for them. Moreover, the driverless car also neglects the last 30 feet (10 mètres) that one will have to cover from curbside to house or services. Therefore, they still have to be thought as a form of public transportation and should be presented as so.

Regarding identification of at-risk drivers and maintenance of one’s license, a clearer portrait of the France license continuum should be made because it is hard to understand why you have 23% of the drivers who present MMSE under 27 (see Table 1), full or average score obtained should be given to better present the sample. From line 238 to 245, solutions should be identified to alleviate this limitation from the results. Moreover, the opening of the paragraph is hypothetical since and should be used with caution since there is no data from the study or from other French study to support it.

Line 266: social interaction (not sociological), change elsewhere

References should be checked to ensure that they are all presented in English (ex. Février, éditeurs, cite, etc.).

As a limitation, it should be noted that the sample is built upon a subsample of a study which aimed at “describe therapeutic management of outpatients”. Therefore, there is higher chance that the sample present medical diagnostic that might not be compatible with driving.

Finally, would there be an opportunity to use some other factors as proxy to identify faster SA? As an example, only using the IADL use of mean of transport ?

Overall, as a clinical perspective, a confusion matrix should be used in order to identify who was or was not classified correctly as drivers and SA (or not).

Reviewer #3: PONE-D-22-26799

“Association between car driving and successful ageing. A cross sectional study on the "S.AGES" cohort” by E. Baudouin and colleagues

The manuscript to be reviewed investigates whether driving status of aged individuals can be considered as a proxy of successful ageing (SA), comprising physical, psychological and social factors. It included non-drivers or drivers aged 65 and above. The study-cohort is part of the larger cross-sectional study S.AGES (Sujets AGÉS—Aged Subjects), an observational prospective cohort study looking at General Practicioner´s (GP) management of patients suffering from three chronical physical conditions, namely arterial fibrillation, chronic pain and diabetes mellitus type 2. Overall, it is scholarly written and informative. The topic is worth investigating.

Nevertheless, I have a couple of comments the authors may consider to specify some aspects to improve their work.

The model of successful ageing (SA) by Rowe and Khan referred to in the study manuscript is important but still one of multiple definitions of successful ageing that do exist (see review in Estebsari F. et al Current Aging Science, 2020, 13, 4-10), and has been refined and extended meanwhile by some authors (e.g. 2002 by Crowther and others). Rowe and Khan have published their model in 1997, not 1987 as mentioned in the text (Rowe JW, Kahn RL. Successful aging. Gerontologist. 1997 Aug;37(4):433-40. doi: 10.1093/geront/37.4.433. PMID: 9279031.) Relevant citations missing in the manuscript. This should be added and outlined.

E.g. Almeida et al. (2006) defined successful aging according to Mini-Mental State Examination (MMSE) as scoring 24 and above, and to Geriatric Depression Scale (GDS) as scoring 5 and below. Hence, the authors of the study under revision rightly used a more restrictive definition with regard to the cognitive status (MMSE <27). Even more so, as the MMSE alone is absolutely insufficient to evaluate a patient’s driving competence.

In contrast, subjects with a GDS score below 10 are regarded as psychologically healthy, whereas Yesavage et al. (1983), author of the GDS, defined less than 5 points as “no depression” and 5-10 points as “mild to moderate” depression. It is well accepted that somatic comorbidity is a risk factor for depression. Both aspects may affect the results of the study. A more restricted threshold for depressive symptoms is preferable, or the authors may have to explain why they decided otherwise.

The study only included patients > 65 yrs. with somatic disorders (chronic pain, diabetes mellitus and atrial fibrillation) not healthy community-dwelling subjects, a selection bias which should be referred to, even if most older people may suffer from one or the other chronic somatic disorder.

It does not become clear how GPs who followed up on the study-participants measured the physiological age being lower, equal or higher to the patients´ chronological age. Did they use objective measures or subjective impressions prone to low interrater reliability. This should be outlined in the text.

Looking at the three sub-cohort´s results and inclusion characteristics it is worthwhile mentioning that atrial fibrillation and type II diabetes do not seem to impact on the driving status (more drivers with this condition than non-drivers), whereas chronic pain is equally distributed in both groups. Is chronic pain an independent risk factor for reduced successful aging and giving up driving?

The three subcomponents of the successful aging model by the Rowe and Khan refined by Young et al 2009 are weighed unequally in this study, as it is dominated by aspects on the physiological component (a combined factor out of 19 items in the comorbidity dimension plus the autonomy dimension, such as ADL and IADL). Whereas very cursory aspects of cognition and mood (MMSE and GDS<10) for the psychological component and even more superficial on the sociological component, (only represented by the question whether the participant lived alone or not, which may have many different reasons without necessarily meaning that a person is socially isolated or not), have been applied.

Unfortunately, a conclusive measure of quality of life such as HRQOL relevant for successful ageing and driving status is missing (Hajek A 2021).

To sum up, the manuscript “Association between car driving and successful ageing. A cross sectional study on the "S.AGES" cohort” by E. Baudouin and colleagues may pose a relevant contribution to the scientific field of successful ageing. It is in line with the current literature supporting the view that driving is a relevant aspect to wellbeing until older age (in industrialised countries). The shortcomings are most likely due to the fact that the study was not primarily designed to answer the question of successful ageing but to look at GPs management of chronic conditions namely diabetes mellitus, chronic pain and atrial fibrillation. A fact that is outlined by the authors in the text and will be evaluated by the editors. If this is considered suitable, after minor revision taking into account the remarks made above a publication in PLOS One can be recommended.

Kind regards and

bonne chance!

6. PLOS authors have the option to publish the peer review history of their article (what does this mean?). If published, this will include your full peer review and any attached files.

Reviewer #1: No

Reviewer #2: No

Reviewer #3: **Yes: **Ute Brüne-Cohrs

---

## [Author Response · Author response to Decision Letter 0]

20 Jan 2023

A rebuttal letter that responds to each point raised by the academic editor and reviewers has been upload as well as a marked-up copy of our manuscript that highlights changes made and an unmarked version of our revised paper without tracked changes. 

Regarding data availibity, due to legal restriction, we are only able to provide detailed descriptive statistics as provided in https://doi.org/10.2515/therapie/2013043, this manuscript its supplementaries but no raw data

The fact that L.B appeared/appears as principal or co-investigator of several clinical trials from different sponsors: Novartis, Actelion, Alnylam Pharmaceuticals, IONIS Pharmaceutical, Takeda, Novonordisk, Shire, Boehringer does not alter our adherence to PLOS ONE policies on sharing data and materials in the competing interest section

---

## [Decision Letter · Decision Letter 1]

5 Apr 2023

PONE-D-22-26799R1Association between car driving and successful ageing. A cross sectional study on the "S.AGES" cohortPLOS ONE

Dear Dr. Baudouln,

Thank you for submitting your manuscript to PLOS ONE. After careful consideration, we feel that it has merit but does not fully meet PLOS ONE’s publication criteria as it currently stands. Therefore, we invite you to submit a revised version of the manuscript that addresses the points raised during the review process.

The manuscript has been responded well by the authors. However, there are still minor comments in the present form. See the suggestions and respond to them appropriately.

We look forward to receiving your revised manuscript.

Kind regards,

Masaki Mogi

Academic Editor

PLOS ONE

Journal Requirements:

Reviewers' comments:

Reviewer's Responses to Questions

**Comments to the Author**

1. If the authors have adequately addressed your comments raised in a previous round of review and you feel that this manuscript is now acceptable for publication, you may indicate that here to bypass the “Comments to the Author” section, enter your conflict of interest statement in the “Confidential to Editor” section, and submit your "Accept" recommendation.

Reviewer #1: (No Response)

Reviewer #2: All comments have been addressed

Reviewer #3: All comments have been addressed

2. Is the manuscript technically sound, and do the data support the conclusions?

Reviewer #1: (No Response)

Reviewer #2: Yes

Reviewer #3: Yes

3. Has the statistical analysis been performed appropriately and rigorously? 

Reviewer #1: (No Response)

Reviewer #2: Yes

Reviewer #3: Yes

4. Have the authors made all data underlying the findings in their manuscript fully available?

Reviewer #1: (No Response)

Reviewer #2: No

Reviewer #3: Yes

5. Is the manuscript presented in an intelligible fashion and written in standard English?

Reviewer #1: (No Response)

Reviewer #2: Yes

Reviewer #3: No

6. Review Comments to the Author

Reviewer #1: (No Response)

Reviewer #2: (No Response)

Reviewer #3: Reviewer 3:

PONE-D-22-26799

“Association between car driving and successful ageing. A cross sectional study on the "S.AGES" cohort” by E. Baudouin and colleagues

The manuscript to be re-reviewed has been revised by the authors and my comments are as follows:

Thank you for considering my comments in general. Nevertheless, I want to outline some shortcomings that are still recommended to be mentioned in the limitations section. As there is the fact that only cursory and superficial aspects of cognition (only MMSE) and the sociological component (living alone or not) have been investigated to relate driving status and successful ageing, reducing the validity of the results. The authors themselves state that a more sophisticated geriatric and cognitive screening is warranted to measure cognitive domains relevant for driving, but without referring to their own study design.

In lines 225-227 the authors state that daily somnolence in the elderly is due to the use of Z-drugs (e.g. Zopiclon, Zolpidem) (ll 255-257). As, in fact, this is not the main and only cause of day time sleepiness in the elderly (neurological and somatic disordes, sleep apnea etc) this should be elucidated or omitted.

The paragraph following this statement (ll 258-264) as well as larger (esp. the revised) parts of the manuscript, are difficult to understand. Hence, proof-reading and language polishing is recommended as the readability of the original manuscript was better than of the revised version.

To sum up, the manuscript “Association between car driving and successful ageing. A cross sectional study on the "S.AGES" cohort” by E. Baudouin and colleagues may still add a relevant aspect to the scientific field of successful ageing. It is in line with the current literature supporting the view that driving is a relevant aspect to wellbeing until older age (in industrialised countries). Still, the shortcomings are most likely due to the fact that the study was not primarily designed to answer the question of successful ageing but to look at GPs management of chronic conditions. This fact will be evaluated by the editors. If this is considered suitable, after some further minor revision a publication in PLOS One can be recommended.

7. PLOS authors have the option to publish the peer review history of their article (what does this mean?). If published, this will include your full peer review and any attached files.

Reviewer #1: **Yes: **Michael Falkenstein

Reviewer #2: No

Reviewer #3: No

---

## [Author Response · Author response to Decision Letter 1]

11 Apr 2023

The reference list have been checked and updated. Every changes since the submission have been noticed and justified for in the rebuttal letter.

The manuscript was proof read by an English native speaker and reviewer's remark have been answered in the rebuttal letter.

---

## [Decision Letter · Decision Letter 2]

17 Apr 2023

PONE-D-22-26799R2Association between car driving and successful ageing. A cross sectional study on the "S.AGES" cohortPLOS ONE

Dear Dr. Baudouin,

Thank you for submitting your manuscript to PLOS ONE. After careful consideration, we feel that it has merit but does not fully meet PLOS ONE’s publication criteria as it currently stands. Therefore, we invite you to submit a revised version of the manuscript that addresses the points raised during the review process.

The reviewer has pointed out some minor spelling corrections.Revise the manuscript according to the Reviewer's suggestions.

We look forward to receiving your revised manuscript.

Kind regards,

Masaki Mogi

Academic Editor

PLOS ONE

Journal Requirements:

Reviewers' comments:

Reviewer's Responses to Questions

**Comments to the Author**

1. If the authors have adequately addressed your comments raised in a previous round of review and you feel that this manuscript is now acceptable for publication, you may indicate that here to bypass the “Comments to the Author” section, enter your conflict of interest statement in the “Confidential to Editor” section, and submit your "Accept" recommendation.

Reviewer #3: All comments have been addressed

2. Is the manuscript technically sound, and do the data support the conclusions?

Reviewer #3: Yes

3. Has the statistical analysis been performed appropriately and rigorously? 

Reviewer #3: Yes

4. Have the authors made all data underlying the findings in their manuscript fully available?

Reviewer #3: Yes

5. Is the manuscript presented in an intelligible fashion and written in standard English?

Reviewer #3: Yes

6. Review Comments to the Author

Reviewer #3: Reviewer 3:

PONE-D-22-26799

“Association between car driving and successful ageing. A cross sectional study on the "S.AGES" cohort” by E. Baudouin and colleagues

The manuscript to be re-reviewed has been revised by the authors and my comments are as follows:

Thank you for going over the manuscript again, as a result the quality of language and readability has improved substantially. Also, my content-related remarks have been considered well.

By going through it again I only came across a few spelling errors or omissions that might be corrected before publication, such as

line 263: ´reach` instead of ´reached`

line 287: `rates´ instead of `rate´

line 289: either past or present tense (see `overestimate´ and `overestimated`)

line 292: `subjects´ instead of `subject´

line 305: `impacted on´

To sum up, the manuscript “Association between car driving and successful ageing. A cross sectional study on the "S.AGES" cohort” by E. Baudouin can be recommended for publication.

7. PLOS authors have the option to publish the peer review history of their article (what does this mean?). If published, this will include your full peer review and any attached files.

Reviewer #3: No

---

## [Author Response · Author response to Decision Letter 2]

17 Apr 2023

All reviewer's remark have been taken into account and manuscript was corrected accordingly. The citations' list have been checked and is complete. No retracted articles have been cited, all changes since submission have been mentioned in the rebuttal letter.

---

## [Editor Report · Decision Letter 3]

20 Apr 2023

Association between car driving and successful ageing. A cross sectional study on the "S.AGES" cohort

PONE-D-22-26799R3

Dear Dr. Baudouin,

We’re pleased to inform you that your manuscript has been judged scientifically suitable for publication and will be formally accepted for publication once it meets all outstanding technical requirements.

Kind regards,

Masaki Mogi

Academic Editor

PLOS ONE
---

## [Editor Report · Acceptance letter]

24 Apr 2023

PONE-D-22-26799R3 

Association between car driving and successful ageing. A cross sectional study on the "S.AGES" cohort 

Dear Dr. Baudouin:

I'm pleased to inform you that your manuscript has been deemed suitable for publication in PLOS ONE. Congratulations! Your manuscript is now with our production department. 

Kind regards, 

on behalf of

Dr. Masaki Mogi 

Academic Editor

PLOS ONE